# Low Cost, High Performance, 16-Channel Microwave Measurement System for Tomographic Applications

**DOI:** 10.3390/s20185436

**Published:** 2020-09-22

**Authors:** Paul Meaney, Alexander Hartov, Timothy Raynolds, Cynthia Davis, Sebastian Richter, Florian Schoenberger, Shireen Geimer, Keith Paulsen

**Affiliations:** 1Thayer School of Engineering, Dartmouth College, Hanover, NH 03755, USA; alex.hartov@dartmouth.edu (A.H.); timothy.raynolds@dartmouth.edu (T.R.); shireen.geimer@dartmouth.edu (S.G.); keith.paulsen@dartmouth.edu (K.P.); 2GE Global Research, Niskayuna, NY 12309, USA; davisc@ge.com; 3German Federal Ministry of Defense, 2E1202 Hamburg, Germany; richter.sebastian@hotmail.de (S.R.); flo3003@web.de (F.S.)

**Keywords:** microwave imaging, breast, multipath, dynamic range, software defined radio, leakage

## Abstract

We have developed a multichannel software defined radio-based transceiver measurement system for use in general microwave tomographic applications. The unit is compact enough to fit conveniently underneath the current illumination tank of the Dartmouth microwave breast imaging system. The system includes 16 channels that can both transmit and receive and it operates from 500 MHz to 2.5 GHz while measuring signals down to −140 dBm. As is the case with multichannel systems, cross-channel leakage is an important specification and must be lower than the noise floors for each receiver. This design exploits the isolation inherent when the individual receivers for each channel are physically separate; however, these challenging specifications require more involved signal isolation techniques at both the system design level and the individual, shielded component level. We describe the isolation design techniques for the critical system elements and demonstrate specification compliance at both the component and system level.

## 1. Introduction

Interest in microwave imaging for medical applications has grown significantly since the early 1980s, but more prominently over the past decade. Much of the recent expansion has resulted from advances in technology that facilitate implementation of novel ultrawideband approaches [1,2,3] and the advent of powerful computing that can accommodate time-consuming inverse problems [4,5,6]. Microwave imaging offers important advantages over other emerging technologies. For instance, microwave methods achieve far greater penetration depths relative to optical imaging techniques, which allow broader applications to be pursued [7,8,9]. Compared to lower frequency electrical impedance approaches, microwave imaging systems provide superior spatial resolution [10,11,12,13]. The list of medical microwave imaging applications is substantial and growing, and currently includes breast cancer imaging [14,15], stroke diagnosis [16,17], bone health assessment [18], and thermal therapy monitoring [19,20,21], among others. In all of these cases, microwave methods either exploit significant tissue dielectric property contrast [16,17,22,23,24,25], or inherent property dependencies [26,27]. While some debate over breast tumor/normal tissue property contrast remains, most studies have demonstrated that substantial contrast exists between tumor properties and those of both the associated adipose and fibroglandular tissue [22,23,24,25,26,27,28,29]. Further, several clinical publications indicate effective tumor detection, diagnosis, and treatment monitoring based on endogenous dielectric property contrast between tumor and normal tissue [14,15,30]. Although the potential of medical microwave imaging is significant, tangible progress in terms of physical implementations has been slow.

Hardware challenges arise from the need to devise a near-field configuration that illuminates the body from multiple directions and receives scattered waves in either monostatic or multistatic modes. In multistatic cases, the difficulties are compounded by antenna mutual coupling, multipath signal distortion within the illumination array, and possible cross-channel leakage within the measurement electronics, themselves [31]. Various strategies have been developed to contend with these signal corruption problems, for example, by multiplexing high-end vector network analyzers (e.g., [32,33]), utilizing lossy coupling media (e.g., [6]), and/or translating small numbers or single source–detector pairs (e.g., [1]).

An alternative and increasingly popular hardware option [34,35] is centered around a measurement system design based on inexpensive, commercially available software defined radios (SDRs). We have described a 4-channel prototype using the B210 USRP boards developed by Ettus Research (Santa Clara, California) [36]. These units incorporate the major functional elements of a sophisticated transmitter and receiver into a single chip (Analog Devices AD9361, Norwood, MA), and operate over an impressively large bandwidth of 70 MHz to 6 GHz. They are finding use in a range of radar applications [34]. Measurement requirements for our log transformed tomographic image reconstruction algorithm along with the associated calibration process have been stable and robust over several generations of hardware systems. Our microwave tomography implementation that does not require a priori information for convergence to the desired solution, and does so with less measurement data relative to competing implementations [37]. These innovations are integral to our success in translating this technology into the clinic [14,15]. Our software advances are described elsewhere, and a comprehensive summary of the most recent innovations appears in Meaney et al. [37]. The system design includes a synchronization strategy, a technique to enhance dynamic range, and a calibration process that compensates for uneven amplitude and phase steps associated with internal variable gain amplifiers [36]. In this paper, we focus on the significant challenges associated with channel-to-channel leakage, packaging SDRs into a compact and light form factor that fits underneath the antenna mounting fixture for our existing microwave breast imaging system, and instrumentation control through existing software platforms. In the case of channel-to-channel leakage, we investigate the problem in terms of (a) leakage through existing transmission lines, and (b) ambient leakage from signals escaping their immediate structures and propagating along the overall structure as surface waves.

In this context, the novel aspects of the system include: (a) enhanced dynamic range, (b) multichannel operation with high degree of channel-to-channel isolation, and (c) data acquisition acceleration via reordering of the data recording sequence. Integration of these attributes enables the realization of an SDR-based acquisition system with data frames rates short enough for use in clinical breast examinations. We demonstrate a compact and efficient packaging of a complete multichannel system, and present results that confirm excellent dynamic range and cross-channel isolation.

## 2. Methods

### 2.1. Overall System Configuration

Figure 1 shows a schematic diagram for a 16 channel system. It includes: (a) a single B210 USRP board operating as a dedicated transmitter, (b) a single-pole, 16 throw switch (SP16T) for multiplexing the transmission channel, (c) a 6-bit, digital attenuator (0−63 dB) for controlling the reference signal amplitude, (d) a series of 16 single-pole, double-throw switches (SPDT) for toggling individual channels between transmit and receive modes, (e) a series of 16 single-pole, single-throw switches (SPST) for extra isolation between the SP16T and SPDT’s (not shown), (f) a series of low noise amplifiers (LNAs) for increasing received signal strength, (g) a series of 16 monopole antennas, (h) eight B210 USRP boards for dedicated reception, (i) a SPST in combination with an 8-way power divider for synchronizing reference signals to each receiver board, and (j) an OctoClock-G (Ettus Research, Santa Clara, CA) for providing coherent clock signals and an accurate reference for B210 signal synthesis. Control, bias lines, USB communication lines, power supply, and controlling computer are not shown. Since our imaging system approach only utilizes transmission data, incorporation of reflection measurements is unnecessary, and would increase system complexity substantially. Substantially greater channel-to-channel isolation was achieved by separating the transmit and receive functions physically and utilizing a dedicated B210 (Ettus Research, Santa Clara, CA) for transmission. The signal from Channel 1 of the transmit board drove individual antennas as each radiated into the illumination tank (illumination tank shown later in Figure 5). Part of the signal was separated via a power divider and passed through a digital attenuator after which it acted as a variable reference signal for synchronization—a process described in more detail in Meaney et al. [36].

Sets of SP16T, SPDT, and SPST switches direct the transmission signal to a single antenna (one at a time) while allowing the reception of response signals by the remaining 15 channels. Transmitting signals must not leak into receiver ports where they will be amplified and detected erroneously by the receive B210 modules. Here, maintaining adequate isolation is challenging because transmitted signals are on the order of 0 dBm while received signals on the order of −140 dBm need to be detected. This level of isolation is uncommon in most measurement systems and demands significant attention. In particular, signal attenuation from transmitters to receivers directly opposing the signal source often exceed −120 dBm and can reach −140 dBm or more at higher frequencies. Lossiness in the coupling liquid confers substantial benefits, specifically, in suppression of surface waves that can corrupt desired signals [31]. Achieving a large dynamic range with concomitant channel-to-channel isolation is challenging but essential for high fidelity data acquisition. As a result, measurement of signals at these low power levels is critical to system success, especially in the breast imaging context. The SP16T switch was purchased in a connectorized configuration from Universal Microwave Components Corporation (UMCC SR-J010-16S, Alexandria, VA) while the SPDT and SPST switches (Peregrine Semiconductor PE4240 and PE4246, San Diego, CA) were integrated into a custom housing (one per channel) with the LNA.

LNAs (Mini-Circuits TSS-53LNB+, Brooklyn, NY) were added to the design to improve the receiver noise figure and dynamic range. The dynamic range of the B210′s by themselves is limited by the internal A/D discretization error at the low end instead of the actual noise level. By adding extra gain in front of these boards, the low end of the dynamic range is determined by the noise floor, which can be controlled by altering the sampling bandwidth—i.e., increasing the sampling time decreases the noise floor. This design approach was discussed in depth in Meaney et al. [36].

The B210 SDRs are driven primarily by the AD9361 (Analog Devices, Norwood, MA) as an agile transceiver, which incorporates two channels of transmission and four ports of reception, respectively (only two of the latter can be used simultaneously). As discussed in detail in Meaney et al. [36], multiport synchronization is possible and leads to a coherent, multichannel system. Variable gain was included in order to measure signals at very low received power depending on sampling bandwidth. In this configuration, Rx ports were utilized for signal reception and one of the Tx/Rx ports was used for reference signal handling. Since the on-board LO oscillators were synchronized, sampling the reference signal on only one of the two remaining ports was required. Furthermore, reference signal sampling could not be performed simultaneously with the signal measurements from the antenna on the same channel, since the Tx/Rx port shared the same pin on the AD9361 chip with the corresponding Rx port. In fact, as discussed in Section 2.5.3., the reference signal was sampled simultaneously with the antenna detection signal from the complementary channel on the same board (only their phase differences were coherent), and then a second scan samples both antenna signals (multiple subtractions produced coherent versions of both antenna—see Section 2.5.3.). In addition, more B210 shielding is critical relative to cross channel leakage, and is discussed in detail below.

The coherent reference signal was fed into a module with three SPSTs placed in series after a digital attenuator that was inserted for extra isolation, which was necessary because the reference signal is not turned OFF during antenna transmission and becomes a potential multipath propagation conduit. The signal was then fed into an 8-way power divider (Pulsar Microwave PS8-12-454/5S, Clifton, NJ) from which the Channel 1 Tx/Rx ports of the receive B210s were driven. Feeding eight receive B210s allowed reception to be performed simultaneously on all 15 channels. The system synchronization utilizing this external reference signal configuration is described in detail in Meaney et al. [36].

The OctoClock-G CDA-2990 (Ettus Research, Santa Clara, CA, USA) provided eight pairs of outputs to the B210 SDR. The first signal is a stable, 10 MHz reference that was used to synchronize the B210s, and the second is a 1 Hz PPS clock signal that was utilized for B210 coordination. In our system configuration, eight signal pairs service nine B210s: one pair is fed into two low frequency power splitters that subsequently drive two B210s.

The list below summarizes the overall system costs. Costs for instruments with comparable performance (such as the Keysight M9800A and Rohde and Schwarz ZNBT) are in excess of $120,000. Utilizing the Ettus B210 SDRs as the foundational building block offers a substantial price advantage.
Ettus OctoClock$2030Ettus B210 (9)$11,538UMCC 1 × 16 Switch$4620Pulsar 1 × 8 Power Divider$750Miscellaneous cables, connectors, and wires$1200Shielded housings$3500Switch/amplifier circuit boards & components (16)$1200NI USB digital I/O boards (3)$370USB hubs (2)$150Total$25,358

### 2.2. Isolation and Shielding

Two components of the hardware require significant shielding to attenuate signals propagating into the environment and back into the system—the B210 USRP circuit boards and the switch/amplifier modules.

#### 2.2.1. B210 USRP Housing Design

B210 USRP circuit boards can be purchased with no shield or one provided by the manufacturer as shown in Figure 2a,b. Our system was comprised of nine B210 boards—one serving as a dedicated transmitter and the remaining eight acting as dedicated receivers (two channels per board to produce a 16 channel system).

In order to develop a suitable shield, we explored several options before converging on the final design shown in Figure 3. An initial design, based on housing walls configured to accommodate existing B210 connectors, was unsuccessful (signals easily escaped through gaps between connector flanges and the housing wall). Instead, coaxial SMA connectors were removed and panel mount flange connectors were utilized to achieve a significantly better seal. The power connector (lower left in Figure 2) was also replaced with a panel mount coaxial cable connector to increase isolation further. The USB3.0 connector was retained on the board and was accommodated through a cutout in the housing wall around its casement.

The shielded housing cover was designed with a raised surface that extended 0.5 mm into the main housing chamber, and fit snugly within the bottom of the housing—adding isolation from ambient fields escaping through seals between housing parts and aligning components to minimize damage during assembly. The board shown in Figure 3 was logically divided into two functional areas: (a) the RF componentry (right) and (b) the digital interface, power supplies, and FPGA controlling unit (left). A gold band was printed on the top surface with plated-through vias to the ground plane below. We incorporated a ridge into the housing cover that closely contacted the gold band to prevent leakage of RF signals from the right portion of the board.

#### 2.2.2. Switch/Amplifier Housing Design

The switch/amplifier module (see Figure 4) incorporated a single-pole/single-throw (SPST) switch (lower left), a single-pole/double-throw (SPDT) switch (lower right), and a low noise amplifier (LNA; top right). The SPDT selects whether a particular antenna operates in transmit or receive mode. In the transmit mode, the signal from the transmit B210 passes through both switches directly to the antenna port for propagation into the illumination tank. In the receive mode, the signal passes from the antenna port, through the SPDT and LNA to the receive port. This SPDT selects the receiver mode, and the LNA added overall gain to the composite receive channel (including the receiver portion of the B210 boards) and reduced the effective channel noise figure [36]. The SPST also added extra isolation so that signals did not leak from the transmitter to the receiver through the circuitry, but instead propagated along the desired path into the tank and through intervening object. The design goal for minimizing leakage from the transmit to receive ports was 80 dB attenuation. The design compartmentalized components such that openings between each acted as quasi-cutoff waveguides, which attenuated unwanted signal propagation between partitions—leaving the coplanar waveguide transmission line as the only viable signal path. Bias and control lines were connected via resin sealed, bolt-in filters (API Technologies Corporation—Spectrum Control—part # 51-729-312, Schwabach, Germany) threaded into the housing floor. Previous analysis of connectorized, single component devices demonstrated that bias and control lines created significant leakage pathways. The extra filtering impacted ON/OFF switching speed; however, elimination of signal corruption was deemed more important even though the design degraded data acquisition speed slightly.

#### 2.2.3. Antenna Mutual Coupling

While not part of the microwave transmit/receive electronics, antenna mutual coupling warrants some discussion. The system feeds an array of 16 monopole antennas submerged in a glycerin: water bath. Figure 5 shows (a) a photograph of a test illumination chamber and (b) a schematic diagram of the imaging field-of-view with a test object present. In our measurement system, a complement of 13 antennas directly opposite the transmitter receives signals from each transmitting antenna. We have studied the effects of the presence of adjacent antennas on the signals received. In Paulsen and Meaney [38] and Meaney et al. [39], we identified the signal perturbations caused by adjacent antennas and developed a strategy for compensating for the effects by modeling each adjacent antenna as an electromagnetic sink. The compensation is effective and produces substantially improved images. Interestingly, at frequencies above 1000 MHz, these perturbations were not observed, primarily because the lossiness of the glycerin-based coupling liquid increased substantially with frequency, and effectively suppressed the interantenna interactions.

### 2.3. Packaging

Key features that directly addressed expansion of our 4-channel prototype [36] were: (1) physical separation of receive modules, (2) separation of the transmit module from receive modules, (3) SPST addition to the switch/amplifier modules to minimize signal leakage from the transmitter when operating as a receiver, and (4) component compartmentalization in the switch/amplifier modules to reduce internal multipath propagation of unwanted surface waves.

Figure 6a,b shows two views of the microwave subsystem as (a) fully assembled and in a (b) separated configuration. The unit was mounted below the antenna array in the operational mode as shown in Figure 6c. Components were organized to minimize wiring and cabling lengths and complexity. For example, groups of eight switch/amp modules had their associated digital I/O cards and power supply screw terminals located directly above to keep bias and control lines as short as possible. Connections to external computers and power supplies were limited to two USB cables, a single 110 V 60 Hz power cable, and two DC power supply wires.

### 2.4. Data Acquisition Sequencing and Time Considerations

Several techniques were combined to accomplish efficient data acquisition: (1) transmission runs continuously, (2) time stabilization occurs at start-up, and (3) transmission antenna switching occurs through the SP16T whose rise time is well under 1 μsec.

The data acquisition sequence was arranged so that a single receive module was selected first and then transmission was incremented over the remaining antennas. Delays because receiver gain levels need to be adjusted did occur (depending on which transmitting antenna was broadcasting), but only involved one or two acquisitions before settling. Here, a single data set was comprised of 10,000 measurements obtained with a sampling rate of 10 MHz for an acquisition time of 8.0 msec. Since stabilization time for each module was identical, the process was initiated for a subsequent module several seconds before the data acquisition was completed from the current module. Data acquisition time for a single receive module from the 14 complementary transmitting antennas and for 5 frequencies required 15 s—total time to acquire data from all 8 receive modules was 2.2 min for a given antenna array position. For each transmitting antenna, a closely coupled channel was always available within the system design, which allowed associated switch/amplifier modules to be connected to the same receive B210 circuit board. Since complete isolation of the transmitting signal from neighboring receive channels was difficult, we omitted the adjacent data points, leaving 14 receive signals for each transmitter (7 boards with two receivers per board). Our previous experience has shown that image quality is not reduced significantly, if at all, when 13 receiver antennas are used instead of the full complement of 15 channels (i.e., the two receivers closest to the transmitter are omitted).

### 2.5. Software and Performance Considerations

Software challenges associated with maintaining data acquisition performance included: (1) retaining coherence between transmit channels on a single board—especially when changing operating frequency, (2) utilizing the receive function on Tx/Rx ports, (3) acquiring signals from different ports on the same board coherently, and (4) delay times associated with each receive B210 board when switching between boards.

#### 2.5.1. Transmit Channel Coherence

To maintain transmit channel coherence, only signals from one of the transmitting ports were used by incorporating a power divider to create two coherent waveforms, and one of which was attenuated digitally and served as the reference signal, which guaranteed coherence (Figure 1). In general, maintaining coherence overcomes limitations that occur when the two transmit signals are not phase locked and different boards, themselves, are not phase locked. Without coherence, the measurement phase becomes arbitrary. We exploited the B210 design feature that the receive channels within a single board are coherent (i.e., differences in their phase are constant), and performed combinations of signal acquisitions involving a transmitter reference signal to maintain coherence between the transmitter board and all receiver boards. The details of this design strategy are discussed in Meaney et al. [36].

#### 2.5.2. Tx/Rx Port Receive Function

Each Ettus B210 board was comprised of two channels—an Rx port for receiving signals and a Tx/Rx port for both transmitting and receiving signals. Signal leakage occurred within the B210 board, itself, from an unconnected Tx/Rx port to its associated Rx port such that a reference signal could be connected to the Tx/Rx port, and sampled on the Rx port. Respective signal power levels were optimized to mitigate cross-channel contamination and associated interference through a calibration process.

#### 2.5.3. Coherent Signal Acquisition Across Same-Board Ports

Receive signals from the two channels (Rx and Tx/Rx) on each board were coherent only when sampled simultaneously. Accordingly, reference signal amplitude was tailored to minimize signal corruption from channel crosstalk.

#### 2.5.4. Set-up Time Minimization

Relative to earlier prototypes [40], data acquisition was reconfigured to loop over receive modules, and then loop over associated transmitting antennas for each module. Transmitter channel selection was performed by an external SP16T switch (not by the B210 boards), which offered rise times on the order of 10 nsec, which were negligible compared to receive board start up times. Only a few receive boards can be operational at any given time without overloading the control computer. Board set up times were minimized by starting each subsequent board before its previous counterpart had completed its data acquisition cycle. Under these conditions, the majority of next-board set up times was performed while data acquisition continued, which allowed sets of data consisting of 5 frequencies and 7 vertical antenna positions to be completed in approximately 15 min per breast.

#### 2.5.5. System Calibration

Our imaging algorithm utilizes measurement data formed as differences between field values acquired when an object is present in the illumination chamber relative to when the tank is empty [40]. The subtraction was performed in terms of the logarithm of field values, and therefore, processed both log magnitudes and phases. It also cancelled phase and amplitude variations associated with the individual antennas, cables, and measurement channels, and preserved measurement coherence. Coherence for this configuration was described and discussed in detail in Meaney et al. [36]. Processing log magnitudes and phases required phase unwrapping, and unwrapping strategies are described in detail in Meaney et al. [37].

## 3. Results

### 3.1. Isolation of Individual B210s

To test isolation characteristics of representative configurations, all B210 coaxial connectors were terminated with shielded matched loads and Channel 1 of each board was programmed to transmit 0 dBm signals at frequencies from 0.9 to 1.7 GHz. Configurations under the test included (a) B210 with no shielding and (b) B210 mounted in the custom shielded housing (see Figure 3). Measurements were recorded at representative locations (Figure 7) to assess shielding benefits, and are reported in Table 1 and Table 2 for transmission from TX/Rx Channel 1 and 2 ports, respectively. An Electro-Metrics (Johnstown, NY, USA) hand-held probe (EM-6992) connected to Keysight Technologies (formerly HP) 8563E spectrum analyzer (26.5 GHz) sensed emissions outside of the housing at selected locations.

Predictably, leakage values were reduced for shielded B210s relative to unshielded boards, and advantages were more pronounced at the higher frequencies. For channel transmissions with no shield, leakage signals were high relatively at measurement points 1–4 compared to 5–8 (see Figure 7), likely due to the RF circuitry being immediately adjacent to these connectors compared to the more distant measurement sites (5–8). For points 1–4, shielded values were uniformly superior. For positions 5–8, leakage was also superior uniformly with shielding for the highest frequency, but more uneven for lower frequencies. Interestingly, instances occurred where unshielded values were nearly identical or slightly better than their shielded counterparts. In cases where shielded and unshielded values were similar, individual measurements were promising (above −94 dBm once).

Predictably, leakage values were reduced for shielded B210s relative to unshielded boards, and advantages were more pronounced at the higher frequencies. For channel transmissions with no shield, leakage signals were high relatively at measurement points 1–4 compared to 5–8 (see Figure 7), likely due to the RF circuitry being immediately adjacent to these connectors compared to the more distant measurement sites (5–8). For points 1–4, shielded values were uniformly superior. For positions 5–8, leakage was also superior uniformly with shielding for the highest frequency, but more uneven for lower frequencies. Interestingly, instances occurred where unshielded values were nearly identical or slightly better than their shielded counterparts. In cases where shielded and unshielded values were similar, individual measurements were promising (going above −94 dBm only once).

### 3.2. Performance of the Switch/Amplifier Module

Figure 8 shows plots of insertion loss (or gain) for cases where (a) signals were transmitted to an antenna, (b) signals were received from an antenna and amplified prior to being measured by the B210 receiver, and (c) leakage signals from a transmitter port to receiver port when the switch/amplifier module operated in the receive mode. The latter leakage signal is also shown when the housing cover was removed (i.e., no shielding). For (a), insertion loss was mild and less than 2.4 dB up to 2 GHz, which is nominally the highest frequency used in the imaging system. The result implies that the SPST and SPDT switches were operating efficiently and power loss was minimal in the system transmission mode. Similarly, gain for signals received by antennas (and directed to receiver boards) was relatively flat and ranged from 20.0 to 21.3 dB with modest monotonically decreasing gain up to 2 GHz. Isolation was greater than 80 dB over the frequency range of interest when the housing enclosure was available, and remained above 70 dB up to 3 GHz. In the unshielded case, values were substantially worse and uneven across the frequency range likely due to excitation of surface wave modes that propagated over the circuit board and housing surfaces and recombined in unpredictable patterns with the desired signals at the output. For the enclosed case, small channels between compartments acted as cutoff waveguides and provided isolation for all modes other than the desired Quasi-TEM transmission line mode that led to well-behaved transfer characteristics.

We also tested housing isolation for ambient signals. Figure 9 shows a SolidWorks rendering of the switch/amplifier module with four associated measurement sites: (a) connected to the antenna (Ant), (b) connected to the transmission network (Tx), (c) connected to the receive B210 boards (Rx), and (d) not directly associated with a connector. In one test, a 0 dBm signal was supplied to the Tx port while the other ports were terminated with shielded 50 ohm loads. Switch control lines were set so that the signal was transmitted to the antenna port. Measurement data are presented in Table 3. Interestingly, no significant difference occurred in shielded and unshielded values for the three frequencies reported in the table. Overall, measurements were lower than −90 dBm. Signals measured near Rx ports were noticeably lower. While signals normally emanating from this port were amplified by the low noise amplifier by about 20 dB, the SPDT switch restricted signal propagation to the port, which more than compensated for the additional gain.

In a second test, a −10 dBm signal was fed into the antenna port while the remaining ports were terminated with shielded matched loads. The SPDT control line was set to direct a signal coming from the antenna port to pass through the low noise amplifier and Rx port. The measurement data are presented in Table 4. Substantial improvement in leakage occurred with the shielded housing relative to the unshielded case for all measurements. Interestingly, signals measured at the Rx site were about 10 dB higher than those from the other sites, which most likely reflected the fact that these signals have been amplified previously by the LNA (which does not occur for the other ports).

### 3.3. System Isolation Specifications

Antenna ports in the switch/amplifier modules were terminated with 50 ohm loads and a 0 dBm signal was the output from the transmit B210 and response signals were measured at each of 16 channels with receiver gain levels set to their maximum (76 dB). Measurements were repeated for frequencies from 700 to 1900 MHz in 200 MHz increments. Figure 10 reports the isolation level measured at each port when the transmit channel was selected to be Channel 1, and remaining ports were activated in the receive mode. Under these conditions, isolation levels were largely below −140 dB and closer to −150 dB. A few measurements for the 700 MHz case rose above the −130 dB level. A few of the 1100 MHz values were also above the −140 dB level.

### 3.4. Images Reconstructed from Measurement Data

For the imaging experiment, we utilized a 2.7 cm × 2.7 cm square cylindrical phantom filled with 60% glycerin and offset 3.2 cm from the center. Its properties were ε_r_ = 51.53 and σ = 1.18 S/m at 1100 MHz. The background medium was an 80% glycerin bath having ε_r_ = 29.87 and σ = 1.25 S/m at 1100 MHz. The height of the liquid was adjusted to be the same as when the object is present and when it was absent. Figure 11a shows measured magnitudes for a single transmitting antenna when the object signals were calibrated, i.e., they had the measurement values for the homogeneous bath subtracted (phases were also shown for completeness). Here, leakage was assumed to arise from any one of the situations described above. Measured data were converted from the log magnitude/phase format to the associated complex numbers. We generated random numbers between −1 and 1 for the real and imaginary leakage signal components and multiplied them by amplitude factors, which were equivalent to those associated with the prescribed leakage simulation levels utilized in the results (in these cases, phases and amplitudes of multipath contributions are unknown. Accordingly, we utilized this random number approach to control the amplitude within a worst-case scenario for the phase. Ultimately, the experiment assesses trends in the presence of progressively distorting signals). These numbers were then added to the complex-valued versions of the original data. Once these terms were summed, they were transformed back to their log magnitude and phase forms, and used directly in the reconstruction algorithm. Here, deviations between the non-leakage case and the background were more pronounced (because these values were normalized to those of the background, the background values were effectively zero). Plots confirm the observations in (a), i.e., that the field values varied most for the most remote antennas. They also show signal deviations increased as leakage levels increased for both magnitude and phase.

Figure 12a reports reconstructed relative permittivity and conductivity images at 1100 MHz with no leakage signal. Images were reconstructed with a Gauss–Newton iterative algorithm under log transformation [37]. Regularization consisted of a standard Levenberg–Marquardt scheme, and the initial image estimate corresponded to the properties of the homogeneous bath. The algorithm completed 50 iterations with an iteration step size of 0.2, which required 5.5 min of computation time to produce each image pair. The permittivity inclusion was recovered at the correct location and with the right size. Artifact levels were minimal in the permittivity images but higher in the conductivity case. In addition, the squared shape of the inclusion cross section is also evident. Conductivity images have the inclusion recovered with the correct size and location, approximately, and the recovered inclusion property values were lower than the background as expected.

Figure 12b–e shows images from the same experiment but with data with increased levels of leakage signals. For the −130 through to −110 dB leakage, the permittivity object was recovered properly but with increasing degrees of artifacts. For the −100 dB case, the image reconstruction diverged. The conductivity object was visible in the −130 dB case, and not discernable at all once signal leakage reached −110 dB. Artifacts increased with increasing leakage signal.

Figure 13a,b shows the plots of permittivity and conductivity values along transects through the center of the imaging zone (the actual property distributions are shown for reference). These results confirm previous observations that the algorithm recovers correct trends in the actual permittivity distribution, although peak values in the object are attenuated because of smoothing inherent in the regularization process. The conductivity profile is flatter as expected because contrast between the object and background is low, and the algorithm recovers values slightly lower than the background in the proper location, even though the object properties are only 5.6% lower than the background. Artifact levels are higher in the conductivity images, which is a consistent feature of our approach as well as other systems [6]. Corresponding plots are also shown for the different leakage levels. These results confirm that when leakage is suppressed sufficiently, quality images can be recovered from the data. As leakage levels increase, image quality progressively decreases.

## 4. Discussion

We examined leakage sources from critical elements in the system—the B210 transceivers and the switch/amplifier modules—because they were the most likely to cause signal contamination. In the case of B210 transceivers, we examined their behavior in the transmission mode while we tested the switch/amplifier modules in both transmit and receive modes. Leakage tests compared instances in which boards were operated with and without the shielding. We also measured leakage effects when boards were isolated with a commercial enclosure supplied by Ettus Research, and these results were not noticeably different from the non-shielded case. Leakage results obtained with custom shields were superior and the leakage signals that did appear were most likely derived from seams between the top and bottom housing parts. While overall system design specifications called for cross-channel isolation greater than 140 dB, the Electro-Metrics probe we used measured levels near −100 dB, which were considered acceptable because these data only reflect a one-way loss through the shielded housing.

Results for the switch/amplifier module exhibited similar trends. Here, we examined both unwanted signals propagating through existing transmission lines, and those caused by surface waves that leaked between shielded housing structures. For the former, leakage signals from the Tx to Rx port were generally below −80 dB. Given this loss would be combined with about 65 dB of attenuation from the SP16T switch associated with the dedicated transmit B210 channel, the maximum unwanted signal reaching the receive B210 was expected to be on the order of −145 dBm, which is acceptable because the lowest level signals getting to the B210 were expected to be on the order of −120 dBm. Corresponding surface wave-related leakage results were similar. Interestingly, leakage did not appear to be significantly better in the shielded case in the transmit mode. Nonetheless, overall leakage levels were low and suggest these signals were not problematic. When the probe was placed above uncovered circuits at random positions, we observed transient values often 30–40 dB higher than those measured near connectors. In the receive mode, the shielded housing offered a clear advantage and isolation data reported here were informative relative to overall system performance.

Field measurement power levels varied when no breast was present relative to when breasts of different sizes and densities were placed in the illumination tank. For example, the magnitude plot in Figure 11 for the case when no object or breast was present represented a lower bound on signal amplitude for the particular liquid and operating frequency given the lossy coupling medium. When the breast was present, signal strengths were higher, and could be as much as 30 dB higher for a fatty breast and 5–10 dB higher for denser breasts. The example emphasizes the fact that central antenna signals were impacted most by leakage levels, similarly to the phantom results, and that the data acquisition needs to handle a broad range of signal levels. Our imaging algorithm was able to recover images of quality similar to those reported in the past for standard phantoms with relatively high property contrast relative to the background. These results were consistent with image reconstruction performance obtained with data acquired with previous hardware implementations of our imaging system, and confirmed the multichannel software defined radio approach described here was operating properly.

While classical imaging defines resolution in terms of the Rayleigh criteria, we were typically able to recover objects considerably smaller than the λ/2 standard. Our experience indicates shows we could detect objects as small as λ/12, although we could not characterize their size or electrical properties, accurately. Experiments with varying levels of noise indicate that resolution was inversely related to the noise level.

## 5. Conclusions

We developed a high performance, multichannel measurement system that could be used for microwave imaging of biological targets. Novel aspects of the system include: (a) an unusually high dynamic range that is critical in medical applications involving lossy coupling media; (b) multichannel SDR-based design with multipath signals reduced to levels having minimal impact on reconstructed image quality; (c) reorganization of data acquisition sequencing to reduce data recording time significantly; and (d) complete integration of these features that makes clinical microwave exams possible. The system operated from 500 MHz to 2.5 GHz, allowed transmission and reception from 16 channels, had a dynamic range of roughly 140 dB, and achieved excellent cross-channel signal isolation. The system design is compact and can be located below the associated antenna array. It also included design features required to ensure coherence between transmit and receive signals. By exploiting emerging technology designed into commercially available software defined radios as a fundamental system building blocks, overall system costs and complexity were reduced dramatically while still leading to a high-performance measurement system.

## Figures and Tables

**Figure 1 sensors-20-05436-f001:**
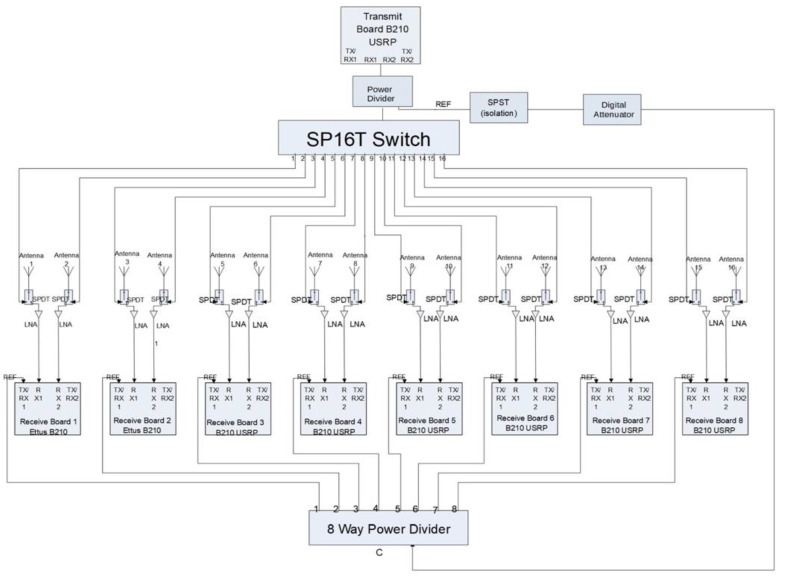
Schematic diagram of the complete system illustrating: (**a**) transmitting B210 board, (**b**) transmitting SP16T switch, (**c**) 16 switch/amplifier modules, (**d**) eight dedicated receive B210s, (**e**) switch module for the reference signal, and (**f**) 1-by-8 power dividers for the reference signal, respectively.

**Figure 2 sensors-20-05436-f002:**
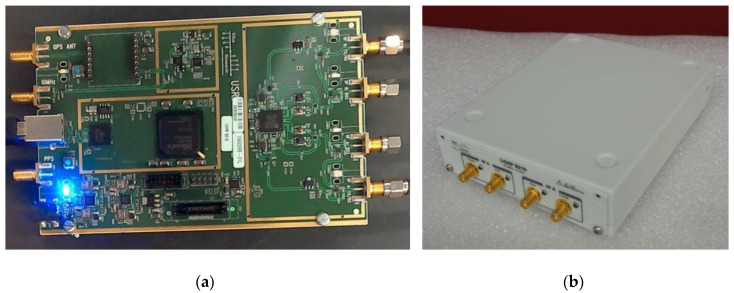
Photographs of the B210 USRP circuit board (**a**) without and (**b**) with a commercial cover.

**Figure 3 sensors-20-05436-f003:**
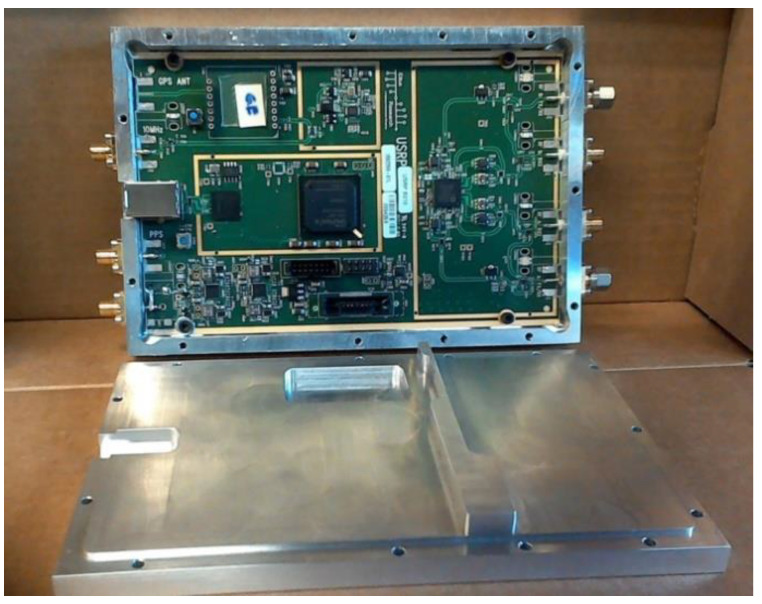
Photograph of the B210 USRP circuit board mounted inside its custom shielded housing and associated cover, which exhibits a central ridge that isolates RF fields from the digital portion of the circuitry.

**Figure 4 sensors-20-05436-f004:**
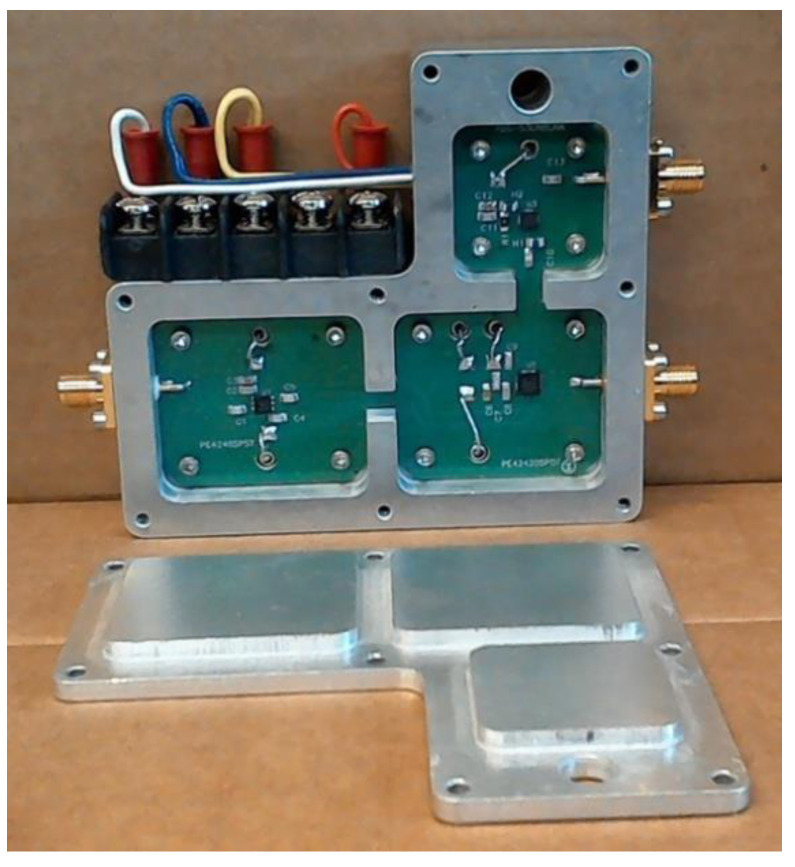
Photograph of the switch/amplifier module illustrating compartmentalization of the single-pole/single-throw (SPST; left), single-pole/double-throw (SPDT; lower right), and low noise amplifier (LNA; upper right), and associated cover with raised surfaces.

**Figure 5 sensors-20-05436-f005:**
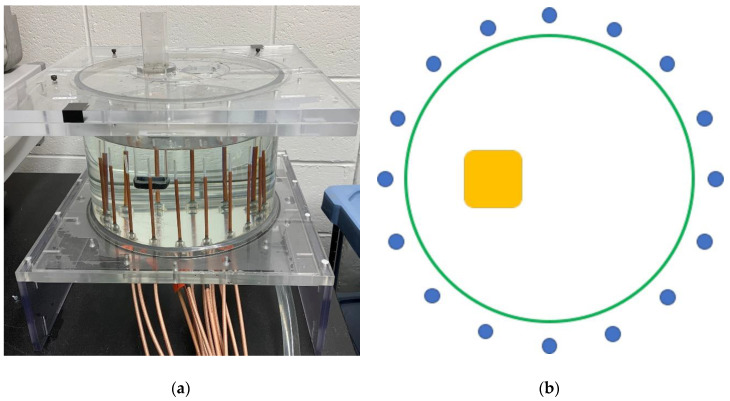
(**a**) Photograph of a test illumination chamber and (**b**) schematic diagram of the imaging field-of-view with 16 monopole antennas and the presence of a yellow test object.

**Figure 6 sensors-20-05436-f006:**
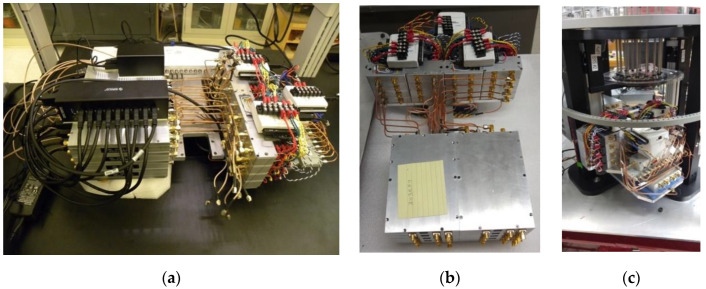
Photographs of the microwave electronic subsystem showing: (**a**) a complete system fully assembled external to the imaging system, (**b**) electronics with the grouping of eight shielded B210s (bottom) and switch/amplifier modules (top; USB hubs removed to expose componentry), respectively, and (**c**) a complete system integrated below the imaging tank and supported by the antenna array mounting plate.

**Figure 7 sensors-20-05436-f007:**
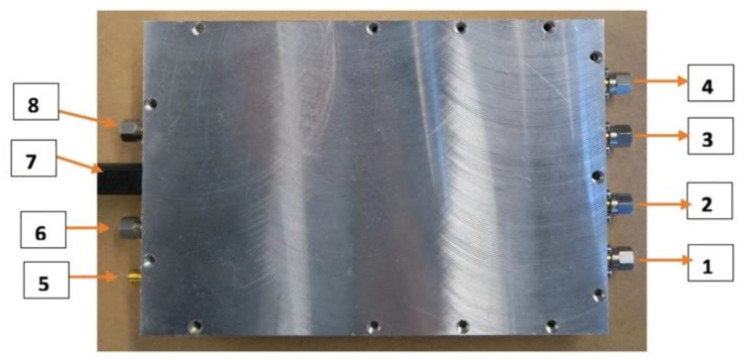
Photograph of the enclosed B210 with labels indicating probe measurement sites.

**Figure 8 sensors-20-05436-f008:**
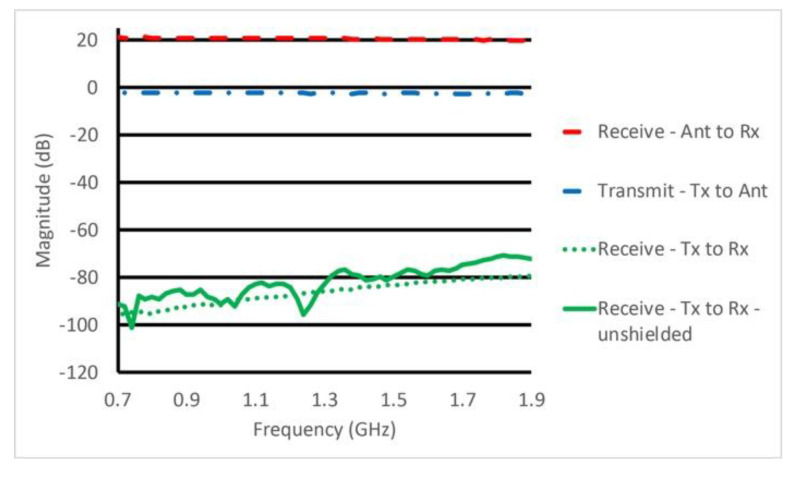
Plots of the switch/amplifier insertion loss (or gain) for the transmission mode (between the Tx and Ant ports), and receive mode (between the Ant and Rx ports). Leakage between transmit and receive ports while operational in the receive mode is shown for the completely shielded housing and the compartmentalized housing without cover, respectively.

**Figure 9 sensors-20-05436-f009:**
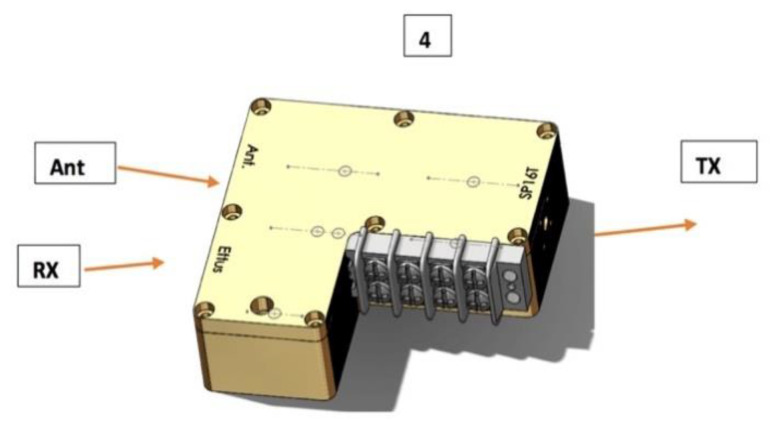
SolidWorks 3D rendering of the switch amplifier housing with four measurement sites.

**Figure 10 sensors-20-05436-f010:**
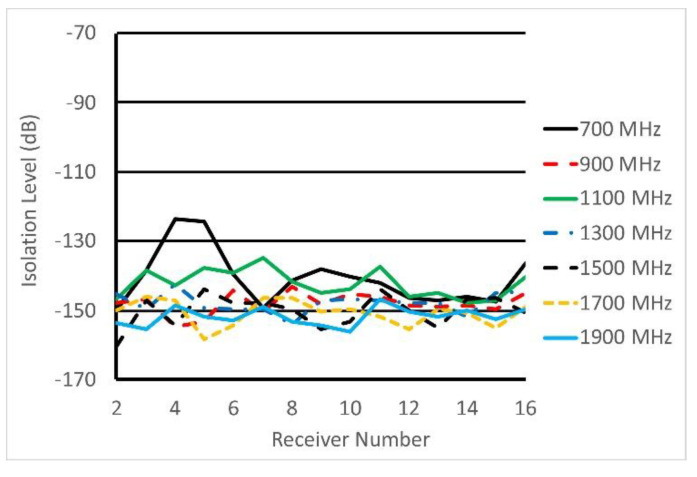
Isolation levels measured at receivers for signal transmission from Channel 1 at 7 frequencies when remaining channels were activated in the receive mode and antenna ports were terminated with a 50 Ω matched load. Except for the 700 MHz case, all values are less than or equal to −135 dB.

**Figure 11 sensors-20-05436-f011:**
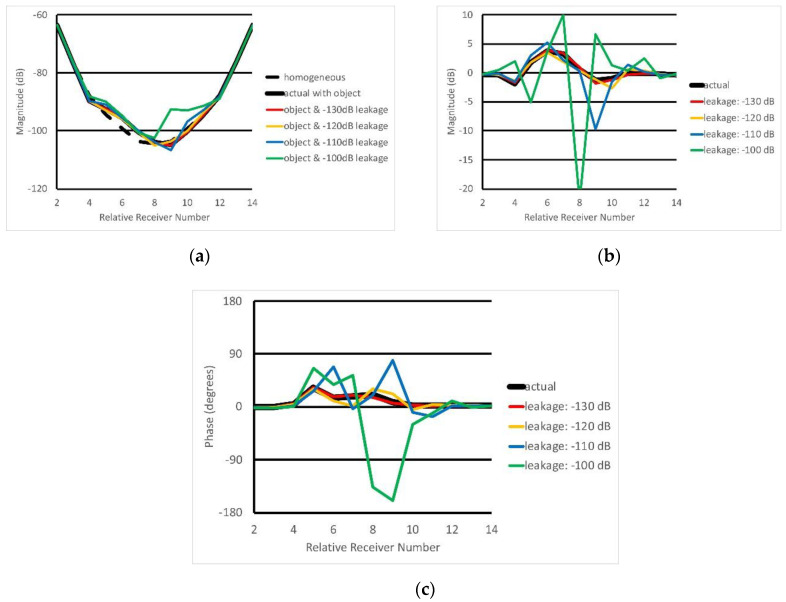
Plots of measurement data for antenna 1 transmission including cases with varying levels of added leakage signal: (**a**) raw magnitude, (**b**) calibrated magnitude, and (**c**) calibrated phases. The dashed black line represents the measured signal for the homogeneous bath whereas the solid black line indicates results when the object is present, respectively. Colored lines symbolize signals when the object is present but with progressively increasing leakage added.

**Figure 12 sensors-20-05436-f012:**
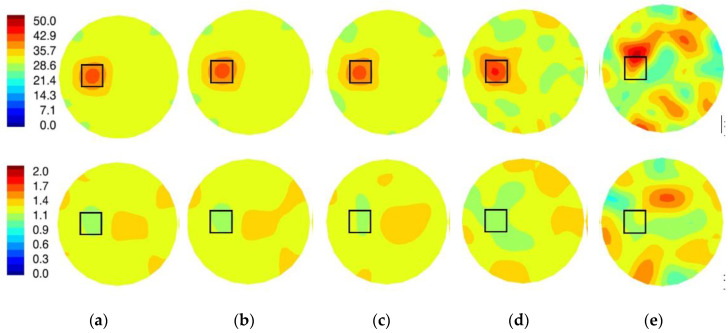
Reconstructed relative permittivity (**top**) and conductivity (**bottom**) at 1100 MHz for (**a**) no signal leakage, and (**b**–**e**) signal leakage of −130 dB, −120 dB, −110 dB, and −100 dB, respectively, of the square object depicted in Figure 5 for the 14.2 cm diameter field of view.

**Figure 13 sensors-20-05436-f013:**
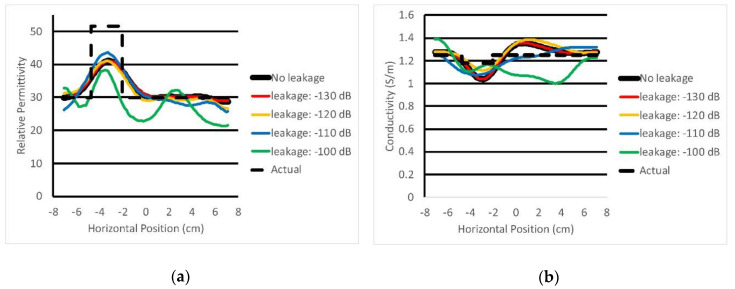
Horizontal transects through the 1100 MHz (**a**) permittivity and (**b**) conductivity images shown in Figure 12.

**Table 1 sensors-20-05436-t001:** Measured leakage signal values at 900, 1300, and 1700 MHz for circuit board locations with: (**a**) no shield and (**b**) shielded housing, respectively, for Channel 1.

Location of Measurement Points	No Shielding (dBm)			Custom Shield (dBm)			Difference (dBm)		
	900 MHz	1300 MHz	1700 MHz	900 MHz	1300 MHz	1700 MHz	900 MHz	1300 MHz	1700 MHz
Location 1	−74	−80	−83	−108	−106	−109	−34	−24	−26
Location 2	−66	−70	−77	−99	−98	−107	−33	−28	−30
Location 3	−86	−75	−74	−112	−111	−107	−26	−36	−33
Location 4	−84	−80	−73	−106	−112	−109	−22	−32	−36
Location 5	−94	−99	−92	−105	−112	−115	−11	−13	−23
Location 6	−94	−109	−94	−94	−102	−112	0	+7	−18
Location 7	−98	−97	−85	−95	−102	−114	+3	−5	−29
Location 8	−100	−92	−78	−101	−106	−114	−1	−14	−36

**Table 2 sensors-20-05436-t002:** Measured leakage signal values at 900, 1300, and 1700 MHz for circuit board locations with: (**a**) no shield and (**b**) shielded housing, respectively, for Channel 2.

Location of Measurement Points	No Shielding (dBm)			Custom Shield (dBm)			Difference (dBm)		
	900 MHz	1300 MHz	1700 MHz	900 MHz	1300 MHz	1700 MHz	900 MHz	1300 MHz	1700 MHz
Location 1	−81	−89	−90	−108	−112	−109	−27	−23	−19
Location 2	−85	−81	−84	−114	−113	−102	−29	−32	−18
Location 3	−82	−78	−80	−96	−99	−102	−14	−21	−22
Location 4	−78	−87	−81	−92	−94	−102	−14	−7	−21
Location 5	−100	−89	−94	−106	−111	−112	−6	−22	−18
Location 6	−97	−89	−102	−99	−108	−114	−2	−19	−12
Location 7	−92	−90	−102	−92	−102	−114	0	−12	−12
Location 8	−90	−88	−92	−99	−107	−116	−9	−19	−24

**Table 3 sensors-20-05436-t003:** Average transmission network (Tx) mode leakage signal measurements at designated locations for 900, 1300, and 1700 MHz, respectively.

Location of Measurement Points in TX Mode	No Shielding (dBm)			Custom Shield (dBm)			Difference (dBm)		
	900 MHz	1300 MHz	1700 MHz	900 MHz	1300 MHz	1700 MHz	900 MHz	1300 MHz	1700 MHz
Ant	−104	−107	−107	−97	−106	−106	+7	+1	+1
Tx	−101	−99	−99	−91	−99	−87	+10	0	+12
Rx	−109	−105	−115	−109	−100	−107	0	+5	+8
Location 4	−95	−90	−92	−97	−95	−103	−2	−5	−11

**Table 4 sensors-20-05436-t004:** Average receive B210 board (Rx) mode leakage signal measurements at designated locations for 900, 1300, and 1700 MHz, respectively.

Location of Measurement Points in TX Mode	No Shielding (dBm)			Custom Shield (dBm)			Difference (dBm)		
	900 MHz	1300 MHz	1700 MHz	900 MHz	1300 MHz	1700 MHz	900 MHz	1300 MHz	1700 MHz
Ant	−91	−87	−96	−115	−116	−112	−24	−29	−16
Tx	−105	−88	−87	−114	−117	−118	−9	−29	−31
Rx	−83	−82	−84	−102	−102	−107	−19	−20	−23
Location 4	−104	−86	−80	−115	−115	−118	−11	−29	−38

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
