# Peer review of "Low Cost, High Performance, 16-Channel Microwave Measurement System for Tomographic Applications"

_sensors, 2020, doi:10.3390/s20185436_

Round 1

Reviewer 1 Report

Can be accepted now. 

Reviewer 2 Report

I feel this manuscript is well written, and covers the extremely important development of more preclinical-based microwave imaging, as opposed to the more basic science oriented approaches of other groups.  The solution of practical problems such as isolation are required if this technology is to move further into the realm of routine human imaging, where it has shown promise for so many years.  I have some minor modifications that I feel would improve the manuscript and increase the enthusiasm with which it will be read by the community.

The first is that, while the results of the isolation adjustments have been covered well, and results are shown on the phantom, I feel that more motivation may be necessary.  While the authors state the necessary isolation system specifications, I believe some statements and justification for the size and fat content of different breasts, and how the necessary isolation would vary with these parameters, would demonstrate the practical improvement achieved by the use of this isolation.  Showing how a 10 dB change in isolation could mean the difference between locating a tumor and it being buried in the noise for certain breast characteristics would justify the added complexity of the isolation presented in the manuscript.

I also feel that for Tables 1,2,3 and 4, a fourth column, showing the difference between no shielding and the shielding at different frequencies and locations would make it much easier at a glance to see the improvement.  Bolding the frequencies and locations with the largest improvement would also keep those specific measurements in mind when reading the results. 

Reviewer 3 Report

This paper presents a measurement system for performing tomographic microwave measurements for breast imaging. The measurement and imaging results show that the system achieves a high dynamic range, and thus would be very suitable for use in microwave imaging applications.

specific comments:

Lines 37-38: “Significant” may be overstating the situation, in some of the studies particularly for breast the contrast is relatively small

Lines 50-68. In general, the motivation for this work and why this advance over the state of the art is needed is unclear. The paragraph notes that this is an increasingly popular hardware choice, but the reason for why is not clear (I am assuming purely the cost?)

Lines 56-57, the authors refer to the log transformed image algorithm -> some additional information would be useful, such as if this is considered a radar algorithm or not and why it has specific promise for microwave imaging. Otherwise the relevance of this sentence is unclear.

Line 94 - refers to an illumination tank, but this hasn’t been described up to this point. The novel system design components should be clearly differentiated from the components of the experimental test set-up.

-140dB value is given is a key target constraint, but it is unclear why this is the aim. Further, a lot of attention is paid to isolation in the circuitry, but it seems like small sources of noise or interference in the circuitry would be overwhelmed by larger noise signals received at the antennas. For example, if the person moves/breathes, the tank liquid is disturbed, differences between empty tank and tank with person, these will all introduce noise.. A discussion of how the 140 dB value is important, based on the clinical scenario, would be useful to include.

Lines 243-244. There are 16 antennas, but I am confused about what the “14 complementary transmitting antennas” refers to here. Should it be 15? I am also confused about the next sentence ‘8 receive modules” - for any given transmitter, are there not 15 receivers?

Section 2.5.1. Please discuss why the channel coherence is important / how the reference signal is used here.

Is Figure 8 measured data? From line 291, it seems that the measured frequency range is 0.9-1.7 GHz, but Fig. 8 shows data across a broader range. Can Table 1 and 2 be updated to include higher frequencies up to 3 GHz. Same question for Figure 10.

Section 3.4. At what frequency are the reported relative permittivity and conductivity values for?

Section 3.4 - when the measurement of the homogeneous tank is done and the object is inserted, is the volume of the glycerin bath reduced to compensate for the volume now occupied by the phantom (i.e., so that the liquid height in the tank is the same for both scans?)

Lines 387-392: It is unclear the purpose of the generated random numbers

In Figure 11 - the black traces are the actual measurements. Are the colored ones synthesized based on the measured data? More description is needed to understand.

Figure 12 - top graphs should be referred to as “Relative permittivity”

-The system (example Fig. 6) solution seems quite bulky, perhaps comparable in size/dimensions to off-the-shelf solutions. It seems that the motivation is to have high-performance with lower costs than commercial VNA/switching matrices.  If this is the primary motivation, then a discussion of the costs of the components used in this system and how they compare to off-the-shelf solutions should be provided.

Round 2

Reviewer 3 Report

The authors have addressed the majority of reviewer's comments satisfactorily.

One question remains, related to the detection of the -140 dBm signal level (page 3, line 16 of the manuscript): Why is this the level that needs to be detected? Some references or justification would be helpful. 

Further, there is a very high number of self-citations. I count at least 18 references to the authors' own works. The authors should check if all of these are actually necessary.
